# Will Macau’s Restaurants Survive or Thrive after Entering the O2O Food Delivery Platform in the COVID-19 Pandemic?

**DOI:** 10.3390/ijerph19095100

**Published:** 2022-04-22

**Authors:** Fan Cheong, Rob Law

**Affiliations:** 1Asia-Pacific Academy of Economics and Management, University of Macau, Macau SAR 999078, China; 2Department of Integrated Resort and Tourism Management, Faculty of Business Administration, Asia-Pacific Academy of Economics and Management, University of Macau, Macau SAR 999078, China; roblaw@um.edu.mo

**Keywords:** COVID-19, restaurant, O2O, customer review, dining experiences

## Abstract

COVID-19 presents a formidable challenge to global tourism. One of the emergency measures adopted by the Macau restaurant industry has been to increase its revenue by joining an online-to-offline (O2O) platform. Nevertheless, are there any risks that follow these opportunities? This article aims to explore whether any risks follow these opportunities, which could extend the literature. Study 1 explores the key factors that customers focus on by analyzing the content of customer reviews published on the Aomi platform through Python. Results show that brand credibility, freshness, and taste remained prominent in the customers’ dining experience. Packaging, delivery quality, and hygiene emerged as new factors due to the COVID-19 pandemic and the popularity of the O2O platform. Customers and staff continued to participate in service interactions through these online channels. Meanwhile, Study 2 contributes to the present understanding of O2O services in restaurants by interviewing catering professionals, and the results highlight how restaurateurs adopt their strategies on O2O platforms.

## 1. Introduction

In response to the COVID-19 outbreak, Macau suspended all non-essential economic and human activities by the end of January 2020 [1], forcing all restaurants to shut down business and sharply reducing their income. During the pandemic, survival has become the top priority of these restaurants. One strategy adopted by these restaurants to maintain their livelihoods has been to cooperate with O2O service giants, such as Aomi and mFood. Despite adopting these strategies, however, the COVID-19 pandemic has pushed many people to the edge. Will these emergency practices make or break those restaurants that have been struggling during the pandemic? Our goal is to build and further expand on the current understanding of the process of change.

The integration of online-to-offline (O2O) platforms introduces third-party factors that restaurants cannot control. For instance, the attitude of a courier can affect the assessment and behavior of customers [2]. Moreover, most customers are attracted to O2O platforms due to the low prices they offer and their ability to compare prices [2]. These platforms may also cause other problems that can harm the reputation of restaurants [3].

The dining experience in restaurants has been extensively investigated in the literature [4,5,6], and O2O-related research has focused on the behavioral intent and evaluation of platforms [7,8]. However, no previous study has investigated dining experiences in Macau in an O2O environment, and the COVID-19 pandemic has changed the thinking and behavior of customers. Therefore, the O2O dining experience in Macau restaurants during the pandemic warrants investigation. In response to this call, this study combines qualitative and quantitative data, a method that produces sound results that are much more convincing than those obtained by using a single method [9]. This approach is also deemed applicable to multiple stakeholders of the restaurant industry, including its customers and professionals. This study is divided into two parts. Study 1 performs content analysis on customer reviews published on the Aomi platform and determines the critical factors of O2O dining experience in restaurants. Study 2 performs in-depth interviews with 22 catering professionals from different restaurants to further understand and triangulate the results from Study 1. The collected data are then explored from an industry perspective to validate and enrich the survey results. The survey results highlight those vital factors that are critical for online and offline channels and highlight some particular elements of the COVID-19 pandemic.

Therefore, the article reported here extends and differs from previous studies in three critical aspects. First, this study explicitly surveys restaurants in Macau and proposes a unique Python approach to uncover customer concerns from online reviews, paying particular attention to which indicators may lead to positive and negative reviews [10]. We also go beyond the items applied in previous research by adding new items that influence customer reviews [11,12]. Second, we examine which social norms guide restaurant personnel in dealing with delivery problems. Third, we supplemented previous studies (i.e., selected qualitative-only studies) by including both qualitative and quantitative results [10,11,12]. This research provides managers with empirical evidence on the drivers of customer quality and articulates opportunities to improve quality service.

## 2. Literature Review

### 2.1. O2O Takeaway

O2O refers to online purchases from brick-and-mortar businesses that consume products or services in physical stores [7,13]. O2O has become a new electric commerce pattern that connects offline services and goods to online sales, marketing, and evaluation. O2O platforms play essential roles in different life scenarios (e.g., car rental, tickets, and takeout) [14]. The O2O market has boomed in Macau over the past six years, with its market size dramatically growing from 500,000 cumulative users in 2018 to 2.8 million in 2020. Moreover, the number of restaurants in Macau has soared to 1600, occupying 90% of the market share. Due to the convenience they bring to both consumers and traders, O2O platforms have proliferated [7].

However, research on O2O has been sporadic and limited thus far. Three main streams have emerged from the limited research. The first stream primarily examines those factors influencing customer ratings of food delivery applications. The attributes of these applications, such as their design, convenience, credibility, and diversity of food choices, significantly affect their perceived value, thereby reinforcing the intention of users to utilize these applications [7,8]. Another important determinant of the credibility of the information processing path is measured by the source and quality of the information [13]. The customers’ performance expectations, alignment with mindfulness, habits, and self-image, and social interaction have also shown positive correlations with their willingness to use these applications [15,16].

The second stream examines those factors that affect restaurant sales on O2O platforms. For top-selling restaurants, both overall ratings and number of reviews are essential to increasing their sales, and delivery service is particularly critical for low-selling restaurants. The adaptive behavior of restaurants has also been studied in the literature, which reveals that promptly adjusting food quality according to customer preferences is crucial in increasing sales [17].

The third stream examines the determinants of the choice of a service provider and the O2O food consumption experience. Customer loyalty is greatly affected by food quality (e.g., taste, performance, variety, and availability of healthy choices), attitude of delivery people, and consistency of orders, packaging, and transport costs. Customers also develop positive attitudes toward their O2O experience when they feel hedonism, price/time savings, comfort in their previous online shopping experience, convenience, and the utility that comes with their purchase [3]. However, further research on O2O platforms is warranted, and no study thus far has investigated the case of restaurants in Macau.

### 2.2. Factors Affecting the Dining Experience

Several factors are significant to the dining experience of customers, including value for money, brand reputation, and quality of service [18,19].

#### 2.2.1. Service Quality

Previous research highlights the significant effects of service quality on dining experience [18]. Quality of service is a multidimensional hierarchical structure comprising three scales, namely quality of results, physical environment, and interaction [20]. First, quality of exchange refers to the customers’ perception toward service delivery or interaction with employees [20,21]. Research on restaurant dining experience suggests that quality of interaction is a combination of problem-solving, professional, and interpersonal skills [22]. Hwang and Ok (2013) [23] used SERVQUAL [24] to measure employees’ empathy, responsiveness, reliability, and tangibles as sub-dimensions. They propose several salient characteristics that all service personnel should possess, such as friendliness, helpfulness, focus on unique needs, knowledge, providing prompts, accurate, professional, and reliable service [6,18,19,25]. Bitner (1992) argues that service workers’ attitudes, behaviors, and skills define the quality of the service provided and ultimately affect customer evaluations of the satisfied [26].

Second, the physical environment is created by people [26] and significantly impacts the behavioral intentions of customers and contributes to their positive sentiment [19]. In general, the quality of the physical environment includes facility aesthetics, seating comfort, ecological conditions, and spatial layout [5,19,23]. When these environmental conditions provide a pleasant atmosphere within a service facility, customers are more likely to exhibit positive behaviors, such as a desire to stay longer and spend more [24].

Third, quality of outcome measures how consumers receive the services and what services are provided [20,21]. Food quality is considered the most crucial factor in the overall customer dining experience [18,22,25,27]. Many characteristics have been used to estimate food quality, including freshness, nutrition, taste, portion size, appearance, menu variety, smell, and temperature [18,23].

#### 2.2.2. Brand Reputation

Brand credibility refers to the willingness (credibility) and ability (expertise) of a brand to consistently deliver on its promises [28,29]. Trust can increase the willingness of customers to purchase a brand and can reduce their uncertainty [29]. Trustworthy brands are positively associated with lower risks and costs and higher perceived quality, thereby increasing the purchase intentions of customers and leading to favorable evaluations [30]. Meanwhile, brand reputation refers to the trust of customers in the credibility, professionalism, and ability of a brand to provide a satisfying dining experience [31].

#### 2.2.3. Value for Money

Price is linked to the credibility of a brand and reflects quality [32]. As an indicator of reputation and quality, premium prices increase the attractiveness of services and products [24,33]. While some restaurants are supposed to be expensive, customers still expect high value or prefer fair prices due to their iconic, hedonistic, or practical nature [4,33]. Previous studies have shown that positive customer reviews of their purchase experiences can result from high value and fair prices [6,19].

## 3. Methodology

### 3.1. Study 1: Comments of Customers on O2O Platforms

#### 3.1.1. Sample

The O2O is the point of contact between brands and consumers [34]. This research analyzes customer reviews published on Aomi, which is the sole official partner of Meituan (the biggest O2O giant in China) in Macau, to measure the performance of restaurants in the country after the COVID-19 outbreak. The customer reviews for 25 restaurants, including the best-selling restaurants located in Taipa and Macau Peninsula, were collected in December 2021. The selected restaurants were initially identified on Aomi. Afterward, a list of reviews for full-service top sellers were extracted and analyzed. Any restaurant without user-generated reviews or with possible manipulations was excluded from the sample. A total of 49,525 reviews for 25 restaurants were eventually collected. These restaurants had Aomi ratings ranging from 7.9 to 9.5 out of 10. All the extracted comments were analyzed, and foreign words were translated to Chinese.

#### 3.1.2. Steps of Content Analysis

An app crawler and Python (Python Software Foundation, Wilmington, DE, USA) were used in the program content analysis to encode data [35]. The content analysis aimed to identify those factors in customer reviews that attracted the most positive and negative attention. A codebook was created based on those factors that affect dining experience as identified from the previous literature (e.g., quality of service, brand reputation, and meal delivery services) [5,19]. Two training meetings were held before the coding. In the first meeting, 50 comments were selected for pre-testing. After introducing the coding protocol, those problems that emerged during the coding process were discussed. In the second meeting, a new item was added based on the pre-test results, and the coding standard was revised based on the differences noted in the first meeting.

Previous research has recommended the use of different strategies for different sentences [36]. Conditional, interrogative, and sarcastic sentences require unique identification strategies. Given that irony is difficult to detect in customer reviews, this study employed the solution proposed by Tsur et al. (2010) for identifying satirical sentences [37].

After two training sessions, Python (Python Software Foundation, Wilmington, DE, USA) was used to analyze and classify the customer reviews according to the revised coding standards. After the first independent coding, the coders discussed the differences in their coding results and then made a final decision. A reliability assessment was carried out if these coders could still not reach a final decision in the second encoding [38]. Cohen’s kappa was used to evaluate inter-rater reliability [39,40]. The kappa coefficient values for the first and second rounds were 0.87 and 0.89, respectively. Kappa values exceeding the recommended threshold (0.70) were considered reliable. 

#### 3.1.3. Coding Standards

The coding standards used in previous studies were compiled. The components of restaurant experience include quality of outcome (texture of mouthfeel, freshness, portion size, taste, menu variety, appearance, and temperature), quality of interaction (assurance, empathy, reliability, and responsiveness), and value for money. Given that this study focuses on O2O settings, quality of the physical environment was ignored. Elements of transport services include speed, spillage, and service attitude. Brand credibility responds to the perceptions of customers toward the professionalism and trustworthiness of a restaurant brand. Overall customer evaluations include loyalty, satisfaction, and quality of experience (repurchase and recommendation intention).

In the second meeting, the new item “texture” under the theme of outcome quality was added to the codebook. Hygiene also emerged as an essential theme that was disregarded in previous dining experience studies.

### 3.2. Study 2: Interview with Restaurant Managers

After discovering the critical drivers of the O2O experience from a customer perspective in Study 1, Study 2 aimed to collect additional information from an industry perspective. Thus far, only little is known about how the emerging O2O business model places high demands on restaurant professionals and what triggers these professionals to use third-party delivery services and join O2O platforms. This study explores the following research questions:

Research question 1. What are the reasons why restaurants provide/do not provide in-house delivery services?

Research question 2. What are the differences before and after the pandemic, and why do such differences exist?

Study 2 was performed to add further information about the O2O services provided by top-selling restaurants in Macau. This study extends the findings of Study 1 from different perspectives and enriches their impact by exploring the costs and benefits of providing third-party delivery services and entering the O2O market, thus adding a new dimension to the present analysis. From January to February 2022, the research team conducted semi-structured interviews with restaurant industry professionals that were selected through convenience sampling. The researchers also asked other questions, such as the platforms that these restaurants joined apart from Aomi, feedback of customers on their O2O services, and their unique O2O food delivery practices during the COVID-19 pandemic. A total of 15 catering managers from different restaurants were contacted, and their demographic information is summarized in Table 1. All interviews were conducted in Cantonese and lasted for 10 to 15 min. Open-ended coding was performed to summarize the basic concepts emerging from the interviews.

## 4. Findings

### 4.1. Study 1

As shown in Table 2, a total of 127,243 comments from customers were extracted, with positive reviews (*n* = 100,920) outnumbering the negative ones (*n* = 26,323). Those positive attributes that may contribute to loyalty, satisfaction, and overall experience were analyzed first. Outcome quality was the most mentioned attribute (*n* = 45,587), followed by delivery (*n* = 6619) and food packaging (*n* = 3065). Meanwhile, the first three themes related to negative reviews were outcome quality (*n* = 11,243), interaction quality (*n* = 1715), and hygiene (*n* = 925), and these themes have been given a new meaning after the COVID-19 outbreak.

#### 4.1.1. Outcome Quality

##### Texture, Portion Size, and Taste

Given that customers attach great importance to taste, this attribute has attracted a high number of positive and negative reviews. Compared with sit-in diners [18,23,41], O2O diners not only overemphasize portion size, but are also fussy about taste. Poor reviews on taste mainly complained about how restaurants failed to meet the high expectations of customers. Apart from general criticism, these reviews also complained about subpar food texture and taste. These complaints were particularly common for discounted items, such as packages or specials. Two example negative comments are as follows:
“*Do not buy this discount food. I know why it is discounted after one bite because the chicken chop is sour*.”

Problems with portion size were prominent among O2O platforms. When they receive an insufficient portion size, customers usually refuse to order food from the same restaurant again given the time and delivery fees. The following comment expresses the negative sentiment of a customer regarding portion size:“*The taste is so-so. The key is that the portion is relatively small and is not enough for girls to eat, which is a bit sad*.”

##### Freshness

Another compelling theme that emerged during the COVID-19 pandemic was freshness. During the pandemic, customers started to pay attention to the freshness of their food out of fear of catching diseases and compromising their immunity. For instance,“*The mung bean paste had a sour smell as soon as it was opened, and it was going bad. I thought maybe I was wrong, and I tried it, and I wanted to spit it out. I do not know why the restaurant still sells it to customers, I am extremely disappointed, and I will never order again*!”

Many netizens also reported having diarrhea and other uncomfortable symptoms after eating the food they ordered. For example, one customer complained,
“*Sea urchins are not fresh when they turn black. After eating, it succumbs. I do not know if it is related to it or not*.”

##### Menu Variety, Presentation, and Temperature

Although temperature and display received much attention in the food literature, these attributes were relatively ignored in O2O reviews given their irrelevance for takeaway. Some customers were also dissatisfied with the variety of food choices, implying that the type of food provided by restaurants was insufficient.
“*I hope the restaurant adds new options and provides a new menu*.”

#### 4.1.2. Interaction Quality

Interaction quality impresses customers in different ways, although suppliers and customers have limited interaction in the O2O food delivery context. Positive reviews (*n* = 733) mainly focused on service failure recovery and reliability, and negative reviews (*n* = 1715) also focused on recovery and reliability.

##### Empathy

Empathy emerged as a key indicator of interactive service quality. A total of 160 comments praised the care and personalized attention given by restaurants, especially during the COVID-19 pandemic. They appreciated these restaurants’ attention to detail and their provision of free food items, such as complimentary beverages and snacks. One positive comment states,“*The small portion of seasonal vegetables that was given for free is quite good and nutritionally balanced. I will order again next time*.”

##### Assurance, Reliability, and Responsiveness

The customers sometimes contacted restaurants for after-sales service on O2O food delivery platforms. A total of 73 reviews gave high ratings for the courtesy, patience, and friendliness of service staff. One comment states,“*The store owner was so patient and polite when I asked for an explanation. I like it*.”

Comments about reactivity were divided. A total of 69 comments expressed satisfaction with enthusiastic service employees who are always ready to solve the problems of their customers. However, 108 comments complained that they were neglected during service interactions. For instance, one customer complained,“*I waited for two hours today, and I have been calling the restaurant to no avail*.”

With regard to reliability, 178 customers mentioned that their selected restaurants provided services accurately. Among the negative comments, 451 mentioned that the restaurant did not provide them disposable tableware or gave them the wrong order, which are considered novice mistakes that should be avoided.

##### Service Failure Recovery

A new topic, “service failure recovery,” was identified given the presence of more service errors in the O2O context. A total of 253 comments mentioned that the responses of restaurants to service failures restored their faith toward these establishments. They also praised the employees for their awareness of the problem and their swift actions. One customer shared,“*Last time I order pork chop, but I have not received it. This time, I explained to the merchant in the remarks, and they made up for their mistake, which was very good*!”

A total of 715 reviews complained that their selected restaurant did not resolve their problem quickly and effectively.

#### 4.1.3. Food Packaging

Although food packaging was not mentioned in the traditional catering literature, this theme attracted much concern in the O2O context. Overall packaging quality is indicative of the value and quality of a restaurant.

##### Solid Packaging

Leakage-free and well-packed packaging is essential in preserving food before consumption. The customers appreciated the use of leak-proof seals and the individual packaging of liquid products and sauces. They also frequently complained about overflows as can be seen in the following review:“*Half of the soup spilled, the tableware was soaked, and a whole piece of wrapping paper was stuck on the chopsticks. It was a bad experience*.”

##### Premium Packaging

Some positive reviews praised the beautiful design and high quality of their food packaging, which the customers believed set their selected restaurant apart from others. One customer commented“*Food is not the point. The point is that the bags delivered are super cute*.”

However, some customers were dissatisfied with the packaging design and quality offered by restaurants as can be seen in the following comment:“*The packaging was not good, and the juice spilled*.”

##### Eco-Packaging

Customers are increasingly becoming more environmentally conscious and prefer green products and services [42]. A total of 30 reviews praised the use of eco-packaging, which they described as advanced, unique, and elegant. One customer commented,“*The packaging bag is gorgeous, and it can be reused for environmental protection.*”

Meanwhile, 180 reviews negatively commented on how restaurants did not pay attention to environmental protection. One customer said,“*Two plastic bags are delivered together, which is not environmentally friendly. The merchant should adjust it*.”

##### Hygienic Packaging

A total of 165 reviews appreciated the use of hygienic containers and the neat packaging of food. One customer commented“*The packaging is clean and hygienic, and gloves are provided for easy consumption*.”

#### 4.1.4. Brand Credibility

A five-star rating indicates the satisfaction of customers with the excellence and quality of a restaurant. Customers also perceive word of mouth as a guarantee for excellence. If their dining experience exceeds their expectations, then these customers perceive that the brand has credibility. Many reviews left the following comment:“*Sure enough, Macau’s No. 1 Sichuan cuisine*.”

Some customers also hinted that branding was their primary focus. For example,“*Great taste, long-lasting brand*.”

By contrast, some customers complained about the poor and mediocre quality of their food, which did not match what they heard from others, hence tarnishing the reputation of a brand. They were also questioning whether a famous restaurant is better than an obscure one.

#### 4.1.5. Delivery

Takeaways are the basic elements of O2O services, and the delivery process includes service methods and attitudes. Most positive reviews praised the timely delivery of their food. Fast delivery helps maintain the freshness and taste of food at times when dining out activities are restricted.

Negative comments about delivery were mainly related to spillage. Customers are generally unable to determine who is responsible for food that was spilled or damaged during delivery. One commenter shared,“*Just a small problem. I do not know if you did not cover it properly or if the rider caused it. When it was delivered, the contents of a whole bag spilled out.*”

Negative comments also pointed toward the poor and untimely delivery of their food. Restaurants often outsource their delivery services to third parties and are therefore unable to guarantee that their food will be delivered on time.

#### 4.1.6. Hygiene

Hygiene was a prominent theme that emerged during the COVID-19 pandemic. Among the positive reviews, customers praised the restaurants and their products for being “hygienic,” “clean,” “safe,” and “carefree.” These customers judged hygiene based on how restaurants pack their food. One commenter shared“*The merchant is prudent enough to separate the sauce from the pho*.”

However, high customer expectations toward hygiene sometimes led to huge disappointments:“*When I opened it, I found that there was no glue separating the soup, and there was no soup inside the soaked rice. The inside and outside of the plastic bag were wet.*”

#### 4.1.7. Value for Money

Customers have different values. Previous studies show that the value for money of O2O services is negatively affected by competition from lower-end restaurants [4,19]. Food quality largely determines the perceptions of customers toward value for money. Some customers ridiculed the difference between quality and price as can be seen in the following comment:“*Limited meat, small portions, fried beef is not worth MOP 58*.”

Takeaway service allows customers to enjoy the sensory delight of delicious food despite the restrictions in dining out activities. This dining experience is considered novel and valuable as expressed in the following comment:“*Yes, I will order again next time. The takeaway staff took the initiative to drive to the company’s door on a rainy day. It was a good experience*.”

#### 4.1.8. Loyalty, Satisfaction, and Quality of Experience

Reviews on loyalty, satisfaction, and quality of experience were generally positive. The negative reviews on this theme mostly criticized how their dining experience failed to meet their expectations. However, these reviews did not say whether they would recommend their selected restaurants to others or repurchase their products. In contrast, those customers who had a satisfactory dining experience expressed their willingness to order again from and recommend their selected restaurants. Their loyalty intentions also extended to different dishes, as can be seen in the following comment:“*The taste is good. I want to try other dishes next time*.”

#### 4.1.9. Visualization of Results

Gephi (Gephi Consortium, Solihull, UK) was used in the network co-occurrence analysis to understand the relationship among the critical attributes before and after the COVID-19 outbreak. Results of the first analysis reveal that taste was strongly related to portion, timeliness, and repurchase intention before the COVID-19 outbreak (Figure 1). Given that the relationships among the critical items may have changed after the pandemic, a secondary analysis was performed based on the comments published after the COVID-19 outbreak (Figure 2). Results show that spillage, reliability, attitude, recommendation, timeliness, and repurchase intention belong to the same group. Taste and timeliness show the strongest relationship, followed by repurchase intention and portion. Freshness, temperature, presentation, and empathy belong to the same group, whereas assurance is related to premium packaging.

#### 4.1.10. Graph results

Rating is another vital factor of customer reviews. Each review has a corresponding rating for each aspect of the customers’ experience. Researchers may use these ratings as literal references and aggregate keywords into a single theme. One approach to visualize change in ratings is to construct a “line chart” as shown in Figure 3. Comments on Aomi held a very negative sentiment toward overall quality as suggested by the blue line in Figure 3, which significantly declined between December 2016 and August 2018. An interactive graph can reveal those events and emotional expressions that trigger an increase or decrease in ratings.

### 4.2. Study 2

#### 4.2.1. Reasons for Providing Third-Party Delivery and in-House Delivery Services

Table 3 lists the primary reasons of restaurants for providing third-party or in-house delivery services. Among the 15 selected restaurants, only 1 provides in-house delivery services, whereas all the other restaurants are planning to adopt third-party delivery as a new business model in the long term due to four factors, namely O2O centralized decision making, trend-oriented motivation, customer-oriented motivation, and economic stimulus. Previous economic research has found that extensive vertical integration [38,39] has the potential to exclude competitors from the market. All interviewees shared their economic motives, including their desire to ease the impact of the pandemic on their businesses. Restaurants have generated most of their profit from takeaway orders during the COVID-19 pandemic. The interviewees also mentioned that they are customer-centric and do not have enough employees to provide delivery services during the pandemic, preventing them from serving their existing customers. These interviewees also shared that they are trying to keep up with the rising trend of mobile shopping and e-commerce, the emergence of food delivery platforms as a global trend, and the Macau government’s promotion of smart tourism. Some of them also mentioned that they make their decisions based on the strategy adopted by O2O platforms (e.g., some takeaway platforms state in their contracts that all takeaway orders from a restaurant should be handed over to the platform for delivery, even if the delivery location is just several floors above the same restaurant).

Some restaurants on Aomi do not use the third-party delivery services for certain reasons. For instance, they may rely on their in-house delivery services or only offer dine-in options to their customers. Third-party delivery services also encounter major issues, such as (a) inconsistency in the food quality guaranteed by the O2O platform and the restaurant (e.g., some drivers deliver their food very late because they take too many orders at the same time), which may affect brand reputation; (b) poor cooperation conditions (e.g., exceeding the cooking time limit would reduce the revenue of the takeaway platform); and (c) additional charges in O2O platforms (e.g., delivery fees).

The interviewees were also concerned about post-delivery food quality (e.g., freshness), delivery timeliness, and repurchase intention. Their economic considerations include the cost of takeaways, such as making additional investments in O2O platform logistics and personnel. They also revealed that during bad weather, the delivery riders on the platform refuse to take orders unless the delivery fees are increased, which would place restaurants at a competitive disadvantage. Some platforms are also unable to deliver food to faraway areas due to geographic restrictions.

#### 4.2.2. Differences before and after the Pandemic

The participating restaurants felt generally positive about joining O2O platforms, which they believed cushioned the negative effects of the COVID-19 crisis on their businesses and helped them retain their loyal customers and reach a wider audience. To further understand the restaurant business, this section highlights several critical differences before and after the COVID-19 outbreak. Among approximately 2700 restaurants operating in Macau, about 1800 have joined O2O platforms, of which 400 have joined after the COVID-19 outbreak.

##### Value for Money

Following the COVID-19 outbreak, customers have become highly price sensitive and reduced their budget for food accordingly. According to Informant 2,“*Before COVID-19, customers ordering through the O2O platform usually spend*
*50 to 60 MOP, while after COVID-19, customers usually spend 20 to 30 MOP. It is estimated that because the economy is not good or some people lost their jobs, so they have to reduce their spending and focus on low-cost consumption.*”

Restaurants often offer special lunch/dinner packages or discounts to cater to these customers. However, some managers said that these offerings reduced their amount of food orders and even led to customer complaints.

##### Outcomes and Interaction Quality

Most interviewees mentioned that the temperature, appearance, and taste of their food would inevitably be affected in the delivery process. Those restaurants that insist on quality avoid third-party delivery mainly for this reason. Informant 13 said that her restaurant refused using third-party delivery services to preserve food quality, which is the core value of the dining experience:“*Customers reported that they had waited for more than an hour for takeout, and the food had become cold after it was delivered, and they complained to us.*”

The interviewees also noted that online customers might complain about insufficient food portions due to the inadequate information provided on O2O platforms. They also admitted that to adapt to the higher costs involved in O2O delivery, reducing food portions was a reasonable measure.

##### Packaging

After the COVID-19 outbreak, restaurants improved how they packaged their food. The interviewees mentioned that they spent much time testing other containers and packaging methods to preserve the quality of their food until it reaches their customers. For instance, Informant 7 said,“*There are certain requirements for takeout boxes used to store food, including the need to meet the three aspects of ‘heat preservation, leakage prevention, and heat resistance,’ and this problem brought a ‘headache’ for restaurants that want to respond to environmental protection initiatives because Macau does not have many choices, and the materials widely used do not meet the environmental requirements.*”

Many interviewees mentioned the importance of preventing leakage in their packaging. Given the movement of drivers during delivery and the placement of food by layers, the requirements for leakage prevention have become more stringent after the COVID-19 outbreak. Respondents 9 and 10 mentioned that most customers respond to environmental protection initiatives and accept and understand the use of paper straws. However, paper lunchboxes cost more than plastic ones (1.3 MOP vs. 0.8 MOP), and bagasse lunchboxes cost even more (2 MOP).

##### Hygiene

The interviewees also stated that the sanitation problems caused by third-party delivery might pose risks to food safety under extreme weather conditions. Informant 1 emphasized the importance of hygiene during the COVID-19 pandemic:“*Each delivery driver will undergo a comprehensive series of trainings to ensure that they fully understand the hygiene guidelines for online ordering and delivery services during the delivery process.*”

All restaurants have strengthened their hygienic practices, for example, by specifying the responsibilities of online food delivery platforms and reviewing the Food Safety Law and Obligations. Some restaurants adopted intensive cleaning and maintenance methods, focused on controlling the temperature and time of their food deliveries, and observed precautions for picking up and receiving food (Informants 3 and 4).

##### Delivery Speed and Service Failure

Restaurants may experience additional delivery failures during the COVID-19 pandemic for three reasons. First, their kitchen workers do not have a new business model and have no prior experience in working during a health crisis, thereby creating additional pressure on these employees. Informant 11 mentioned that in the first few weeks following the outbreak, they lacked clear directions on when to coordinate with their takeout staff and even ignored the notes left by their customers on O2O platforms during their peak hours, resulting in handover errors and customer complaints. Second, these restaurants required additional staff to package their food, and the lack of human resources may lead to errors and complaints from customers. Third, the COVID-19 pandemic significantly increased the volume of orders on O2O platforms, and these platforms emphasize the importance of express delivery, increasing the changes for restaurants that can lead to service failures (e.g., loss of tableware or incorrect delivery).

However, not all restaurants can provide timely solutions to these problems. According to Informants 12 and 14, they would contact their customers who left terrible reviews to offer them compensation as soon as possible, and their PR team would engage in fault recovery. Meanwhile, Informant 15 shared that her restaurant would improve its processes accordingly to ensure the quality of its food:“*We were originally a tourist business; the taste of our food, our products, and even the atmosphere of our restaurant are actually designed to attract tourists. But now, we can only rely on sporadic neighborhoods to help, so we had to change the taste of our food*.”

Informant 6 admitted that her restaurant did not pay much attention to customer comments posted on the Aomi platform. However, the differences in the attitudes of restaurants toward service failures can significantly influence the repurchase intentions and comments of their customers.

## 5. Conclusions

While the COVID-19 pandemic had devastating effects on businesses, a few industries were still able to go against the current [15]. Local food delivery platforms consistently earned higher profits during the pandemic. Given the growing popularity and use of O2O in the travel industry, understanding the impact on consumer responses to the service experience is critical.

Despite the large number of recent studies examining O2O services [15], there is a lack of focus on why restaurants offer or do not offer in-house delivery services. The current research contributes to the study of existing O2O services on customer satisfaction.

### 5.1. Theoretical Significance

The findings from the two studies identified service quality, cost effectiveness, and brand reputation as bright spots in the O2O scenario [19,31]. Other interesting findings were obtained after further investigating the themes and items expressed in customer reviews. First, the themes of quality (e.g., temperature, freshness, and taste of food) were just as relevant in takeaway services [18,23], whereas texture, hygiene, and portion were more important in O2O dining than in restaurant dining. Second, the quality of interactions between customers and restaurant staff plays a vital role in the O2O context. The importance of empathy was highlighted given the negative effects of the pandemic. Customers also considered their interactions with delivery personnel when evaluating the services they received. Third, food packaging was often mentioned in customer reviews because these customers could not directly experience the atmosphere, decoration, and other environmental clues of restaurants when ordering their food via O2O platforms.

#### 5.1.1. O2O Inducing Factors

The service attitude of the O2O delivery staff affects the dining experience of customers. Specifically, these customers anticipate O2O platforms to provide a convenient, fast, and accurate service [3]. Therefore, a delivery quality that fails to meet these expectations will lead to customer dissatisfaction and is considered a sanitary indicator. However, the delivery outcome is entirely beyond the control of restaurants. Accordingly, restaurant operators are carefully considering the benefits and risks of outsourcing their deliveries to O2O platforms and are presently formulating follow-up plans.

Packaging emerged as a critical environmental cue for restaurant customers [19]. From a marketing perspective, takeaway packaging reflects the quality and value of a brand. Therefore, packaging, including its design, text, material, and logo, can be seen as an environmental cue that affects the emotions and behaviors of customers. First, high-quality packaging may reflect the excellence of a brand [43]. Visual cues not only enhance customers’ perceptions toward the reputation of a brand but also help them understand the symbolic meaning of this brand. Second, storing food in neat and clean packaging makes consumers think that their food is safe and hygienic [44]. Third, with the increasing number of O2O food orders during the COVID-19 pandemic, the environmental impact and sustainability of food packaging have also attracted much concern. Eco-packaging makes people believe in the social and ecological responsibilities of a brand, thus producing positive emotions, especially among high-income customers, as they care more about the environment [45].

The role of packaging is to protect food. Sturdy packaging ensures that the food remains in good condition until it is received by the customer.

#### 5.1.2. O2O Influence Factors

Portion is the second most frequently mentioned factor affecting outcome quality. Some elements in the O2O environment, including portion, did not receive much consideration in previous catering research [23,41]. By highlighting the importance of reasonable food portions, the results of this work fill in the gaps in the literature [3]. The increasing importance of food portions reflects the basic physiological needs of customers during the COVID-19 pandemic, and this is especially true for O2O platforms. Consumer experience is mainly affected by utilitarian values. If customers lack sufficient information about the portion of food, then they may ask servers for their advice before ordering their meals. Otherwise, they may leave complaints.

The once crucial attribute of “food quality” was given less importance in O2O reviews. As customers may be fully aware that the quality of their food may be reduced during the delivery process, they will not develop high expectations for this aspect even if they value how their food is presented during their dine-in experiences [19,23]. Even if their takeaway lacks a delicate presentation, these customers still expect their food to arrive intact.

Reassurance, empathy, reliability, and responsiveness continue to impress customers in the food delivery setting even though O2O takeaways eliminate physical interactions between customers and service staff. For instance, customers will feel at ease when their requirements are taken seriously [27]. The category of service failure recovery received the most negative reviews. Customers become unhappy and dissatisfied when facing unreliable and unresponsive restaurant staff. Their negative reviews mainly revolved around restaurant staff failing to meet their special requirements or to compensate them for getting their orders wrong. Therefore, proper and timely fault recovery services have become necessary during the COVID-19 pandemic. Providing compensation can effectively increase customer satisfaction and promote a mutually beneficial behavior [46].

#### 5.1.3. COVID-19 Influence Factors

The social and government restrictions implemented during the COVID-19 pandemic have greatly limited the interactions and activities among people, forcing people to order food via O2O platforms [47]. When ordering food through these platforms, customers are mostly concerned about the empathy of service staff and the hygiene of their food.

When ordering food via O2O platforms, customers regard hygiene as their top priority and seek risk-free and safe food. People have started paying much attention to cleanliness and hygiene during the pandemic. Low-end restaurants have long been blamed for using expired food, reusing their oils, and operating in unsanitary environments [48]. In contrast, the food provided by mid-to-high-end restaurants is considered hygienic and guaranteed. The careful and clean packaging of food is generally interpreted as a symbol of professional hygiene.

The act of delivering meals in emergencies is interpreted as a sign of sympathy. Those restaurants showing empathy genuinely care about the interests and welfare of their customers. For example, some restaurants include sanitary wipes or sticky notes with inspirational words in their takeout packages [6,19]. The results of this work also emphasize the importance of the attention paid by restaurants to the emotional needs of their customers during a crisis. The challenges brought on by the COVID-19 pandemic have increased the anxiety, stress, uncertainty, worry, and fear felt by people. Under these situations, customers especially appreciate how restaurants pay attention to their emotional needs by showing kind gestures and words.

#### 5.1.4. Cross-Context Dominant Factors

Given that food taste, brand credibility, temperature, and freshness are core attributes that determine the overall dining experience of customers, they are often mentioned in positive reviews [18,23]. Meanwhile, the appearance of these attributes in negative reviews pointed toward the positioning problems of restaurants.

As more restaurants join the O2O market, a positioning paradox started to appear in customer reviews. Many customers were complaining that the taste of their food is not unique enough. Meanwhile, those reviews criticizing the failure of restaurants to provide food as expected may damage the brand reputation of these restaurants. The complaints related to special or discounted products have become more intense during the pandemic. Lower prices have also been associated with poor promotion [4]. Preferential prices are particularly inviting for low-income customers who often hold unfavorable attitudes toward restaurants [49]. The same finding was observed in research on value for money; specifically, the expectations and experiences of individuals influence their perception toward value for money. Low-income customers are accustomed to relying on low-end dining experiences as a reference when evaluating the products or services they receive. These issues, especially the reluctance of restaurants to reduce their costs, were confirmed by the interview data.

### 5.2. Management Significance

People have adopted new ways of living, working, and interacting with others following the COVID-19 outbreak [50]. The increasing use of modern technologies during the pandemic has also triggered a significant increase in the use of O2O food delivery services. To breeze through the negative impacts of the pandemic on their business, restaurants need to incorporate the O2O business model into their long-term strategies. This research offers some practical suggestions for restaurants that are eager to provide their customers with a dining experience through O2O platforms.

First, restaurants need to reach the right customers through the O2O platform to gain a firm foothold in the market segment. Food quality and premium are two important clues that affect the perception of customers toward restaurants [4]. Therefore, the primary agenda of restaurants is to continue ensuring the taste and composition of their food by improving the quality and freshness of their food. These restaurants should also avoid price competitions to maintain their image.

In the O2O environment, customers evaluate their impression toward and consumption experience with restaurants based on their food packaging and express delivery performance. Food packages can set a restaurant apart from others, especially when they are made of high-quality materials and have exquisite designs or logos. Restaurants need to package their food separately in appropriate containers to avoid spills and to maintain the appearance and hygiene of their food. Restaurants may also cater to environmental initiatives by using compostable, durable, and eco-friendly materials in their packaging.

Restaurants should also cooperate with O2O platforms and strive for better control over delivery quality given its importance in the O2O process. Although they outsource their takeaway services to O2O companies, restaurants need to actively pay attention to customer reviews and follow up on negative reviews in collaboration with O2O companies. They may recruit or train employees who specialize in such tasks. They may also work with O2O platforms in formulating hygiene standards, estimating delivery time, and improving the service attitudes of couriers.

O2O takeaway is vastly different from the usual dine-in services in terms of the challenges encountered when providing services to customers. Restaurants should therefore set up dedicated teams to handle customer complaints and inquiries. They should also provide relevant training for their employees and establish a comprehensive standard operating procedure for O2O takeaways. They may also provide their customers some small gifts and greeting cards.

While restaurants in Macau joined O2O food delivery platforms to survive throughout the pandemic, they were generally unsure whether these platforms could help them maintain their good reputation and replicate the dine-in experiences of their customers [51]. This research, therefore, explores the O2O dining experience of customers by analyzing the comments they leave on the largest takeaway platform in Macau and by conducting in-depth interviews with catering professionals. By highlighting the critical factors of O2O consumption and the special comments left by customers in the context of the COVID-19 pandemic, the findings of this work can help restaurant operators formulate strategies for achieving satisfactory results when using O2O platforms [51].

The limitations of this work should be noted when interpreting its findings. First, this study focuses on the services of restaurants from the O2O delivery perspective. While many restaurants have adopted third-party platforms to receive takeout orders, customers may prefer those restaurants that recruit in-house delivery fleets and have dedicated platforms (e.g., WeChat and livestream) for taking orders. Future research may examine the difference between these two patterns from the customer perspective. Second, the quantitative results of this study are graphical representations based on customer ratings of 25 restaurants. More restaurants’ reviews could be included to corroborate the findings and increase the depth of the investigation. As analytical techniques mature, future research may also explore customer ratings by using quantitative methods. Third, future research may use multi-source heterogeneous data (e.g., photos and videos) to increase the depth of their investigation in the age of big data.

## Figures and Tables

**Figure 1 ijerph-19-05100-f001:**
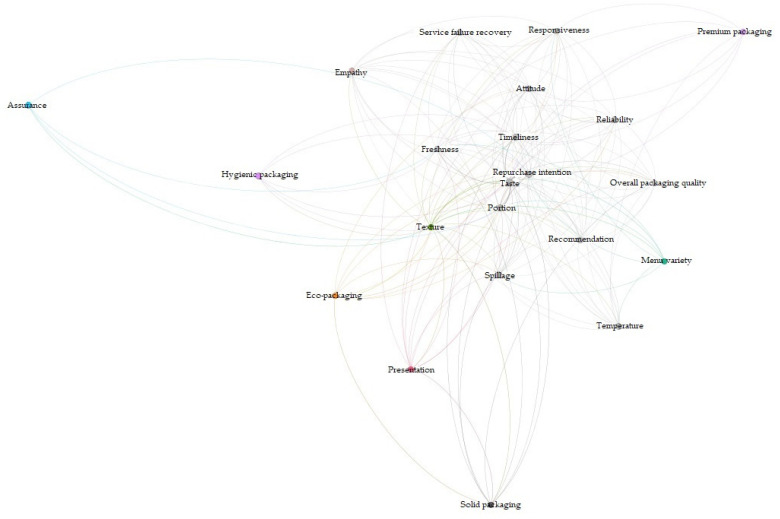
Network visualization of customer comments on Aomi (before COVID-19).

**Figure 2 ijerph-19-05100-f002:**
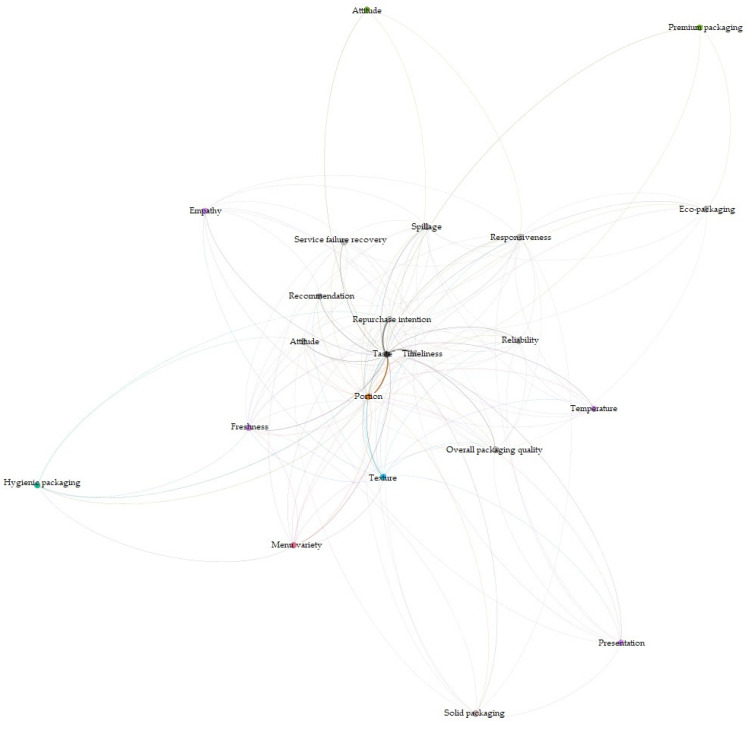
Network visualization of customer comments on Aomi (after COVID-19).

**Figure 3 ijerph-19-05100-f003:**
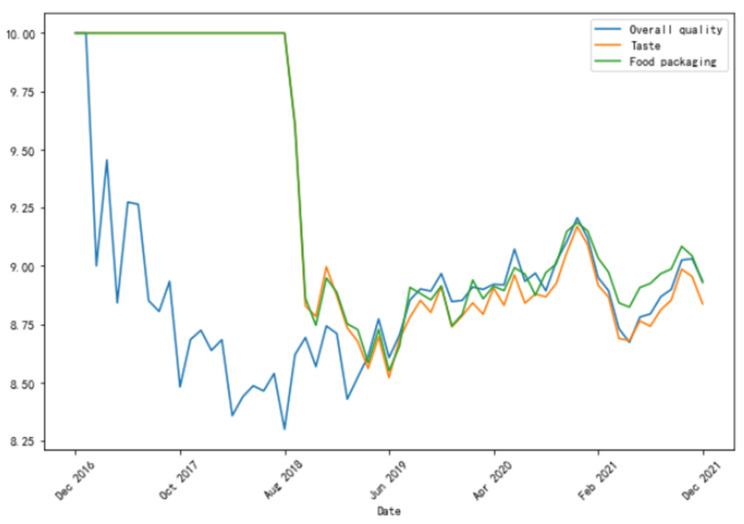
Customer ratings.

**Table 1 ijerph-19-05100-t001:** Socio-demographic background of restaurant managers.

No.	Gender	Age	Tenure in the Catering Industry
1	Female	45	24
2	Female	58	37
3	Female	42	21
4	Female	48	27
5	Female	37	16
6	Female	35	14
7	Male	38	17
8	Male	39	18
9	Male	41	20
10	Female	40	19
11	Female	51	30
12	Male	44	23
13	Female	34	13
14	Female	36	15
15	Female	32	11

**Table 2 ijerph-19-05100-t002:** Items and frequency of customers’ O2O platform reviews.

Themes	Items	Positive	Subtotal	Negative	Subtotal
Outcome quality	Texture	2035 (1.59%)	45,587 (35.82%)	441 (0.34%)	11,243 (8.83%)
	Portion	5658 (4.44%)		3048 (2.39%)	
	Taste	30,122 (23.76%)		4694 (3.68%)	
	Freshness	5267 (4.13%)		431 (0.33%)	
	Menu variety	1324 (1.04%)		17 (0.01%)	
	Presentation	500 (0.39%)		825 (0.64%)	
	Temperature	681 (0.12%)		1787 (1.40%)	
Interaction quality	Empathy	160 (0.12%)	733 (0.57%)	252 (0.19%)	1715 (1.34%)
	Assurance	73 (0.05%)		189 (0.14%)	
	Responsiveness	69 (0.05%)		108 (0.08%)	
	Reliability	178 (0.13%)		451 (0.35%)	
	Service failure recovery	253 (0.19%)		715 (0.56%)	
Food packaging	Overall packaging quality	2699 (2.12%)	3065 (2.40%)	335 (0.26%)	762 (0.59%)
	Eco-packaging	30 (0.02%)		180 (0.14%)	
	Solid packaging	31 (0.02%)		118 (0.09%)	
	Premium packaging	140 (0.11%)		59 (0.04%)	
	Hygienic packaging	165 (0.12%)		70 (0.05%)	
Brand credibility		2053 (1.61%)	2053 (1.61%)	916 (0.71%)	916 (0.71%)
Delivery	Timeliness	4980 (1.51%)	6619 (4.80%)	929 (0.98%)	2615 (0.02%)
	Attitude	1103 (1.24%)		1256 (0.39%)	
	Spillage	36 (0.26%)		430 (1.44%)	
Hygiene		64 (0.05%)	64 (0.05%)	925 (0.72%)	925 (0.72%)
Value for money		1778 (1.39%)	1778 (1.39%)	761 (0.59%)	761 (0.59%)
Quality of experience		6120 (4.80%)	6120 (4.80%)	1387 (1.09%)	1387 (1.09%)
Satisfaction		25,714 (20.20%)	25,714 (20.20%)	3446 (2.70%)	3446 (2.70%)
Loyalty	Recommendation	4953 (3.89%)		1466 (1.15%)	
	Repurchase intention	4734 (3.72%)	9687 (7.61%)	1087 (0.85%)	2553 (2.00%)
Total		10,0920 (79.31%)		26,323 (20.69%)	

**Table 3 ijerph-19-05100-t003:** Reasons for providing third-party and in-house delivery services.

Provide Third-Party Delivery Services	Provide in-House Delivery Services
Trend-oriented motive	Brand concerns
Customer-oriented motive	Economic concerns (additional fees for third-party delivery services via O2O partners)
Economic motivation	Collaboration concerns
Centralized decision making of O2O	

## Data Availability

Data was obtained from Aomi and are available from the authors with the permission of Aomi.

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
