# Peer review of "Will Macau’s Restaurants Survive or Thrive after Entering the O2O Food Delivery Platform in the COVID-19 Pandemic?"

_ijerph, 2022, doi:10.3390/ijerph19095100_

Round 1

Reviewer 1 Report

A document with the revision is attached.

Reviewer 2 Report

This is an interesting paper but needs a revison.

  1. The aim of the article should be clearly stated in the Introduction and the Abstract.
  2. The wording "due to the combination of the COVID-19 pandemic and the O2O platform" has to be changed (Line 13).
  3. The discussion part could be improved.
  4. I suggest to cite more papers on the impacts of COVID-19 pandemic on the economies and the market actors, e.g. this statement needs to be supported by the literature: "While the COVID-19 pandemic had devastating effects on businesses, a few industries were still able to go against the current."
  5. The author(s) should suggest what can be the object of further studies.
  6. References have not been adjusted yet to the requirements of the IJERPH.

Reviewer 3 Report

The paper is interesting and well written. My only concern regards the fact that results refer to a limited number of restaurants.

Reviewer 4 Report

Dear authors,

You approached a very actual topic and your results are useful for the industry. The challenges that restaurants had to face during the pandemic were new and changed the way to do business. Also, the methodology you used is adequate for such a research. Please consider the fallowing suggestions to improve the article.

  • include in the abstract about the research method you used
  • you specified that ”this study combines qualitative and quantitative data”. However, as you described the two methods, they are both qualitative (interview is a qualitative research method and data analysis is also a qualitative research). Please modify accordingly 
  • please update the in -text citation according to the journal recommendations 

All the best!

Reviewer 5 Report

The introduction lacks an argument showing clear and convincingly the importance of the chosen topic, the need to study it and the positioning of the article in the field of knowledge. These tasks require a synthetic review of the literature in support of a compelling argument that your article meets a research need and will make a contribution to the field of knowledge in question. The following publications may be helpful in this regard: Alvesson, M., & Sandberg, J. (2011). Generating research questions through problematization. Academy of management review, 36(2), 247-271. and Sandberg, J., & Alvesson, M. (2011). Ways of constructing research questions: gap-spotting or problematization?. Organization, 18(1), 23-44.

I think that this very current paper, which makes use of User-generated content, can contribute to the results and, in particular, to strengthen the collection methodology: "A user-generated content analysis on the quality of restaurants using the TOURQUAL model" https: //digitalcommons.usf.edu/globe/vol7/iss1/1/

The work lacks a more robust conclusions section, supported by the confrontation of the results with the elements of the theoretical foundation section and that explains relevant contributions to the field of knowledge, as well as recommendations for future studies.

Author Response

This manuscript is a resubmission of an earlier submission. The following is a list of the peer review reports and author responses from that submission.